# Fungal Host Affects Photosynthesis in a Lichen Holobiont

**DOI:** 10.3390/jof8121267

**Published:** 2022-11-30

**Authors:** Meike Schulz, Imke Schmitt, Daniel Weber, Francesco Dal Grande

**Affiliations:** 1Senckenberg Biodiversity and Climate Research Centre (SbiK-F), 60486 Frankfurt, Germany; 2Goethe University Frankfurt, Institute of Ecology, Diversity and Evolution, 60438 Frankfurt, Germany; 3Phytoprove Plant Analytics UG, 60486 Frankfurt, Germany; 4Department of Biology, University of Padova, 35121 Padua, Italy

**Keywords:** photosynthetic performance, lichen-forming fungi, hologenome, fungal–algal pairings, symbiotic mosaic

## Abstract

Corals and lichens are iconic examples of photosynthetic holobionts, i.e., ecological and evolutionary units resulting from the tightly integrated association of algae and prokaryotic microbiota with animal or fungal hosts, respectively. While the role of the coral host in modulating photosynthesis has been clarified to a large extent in coral holobionts, the role of the fungal host in this regard is far less understood. Here, we address this question by taking advantage of the recent discovery of highly specific fungal–algal pairings corresponding to climatically adapted ecotypes of the lichen-forming genus *Umbilicaria*. Specifically, we compared chlorophyll a fluorescence kinetics among lichen thalli consisting of different fungal–algal combinations. We show that photosynthetic performance in these lichens is not only driven by algal genotype, but also by fungal host species identity and intra-host genotype. These findings shed new light on the closely intertwined physiological processes of fungal and algal partners in the lichen symbiosis. Indeed, the specific combinations of fungal and algal genotypes within a lichen individual—and the resulting combined functional phenotype—can be regarded as a response to the environment. Our findings suggest that characterizing the genetic composition of both eukaryotic partners is an important complimentary step to understand and predict the lichen holobiont’s responses to environmental change.

## 1. Introduction

According to holobiont theory, the overall fitness of each organism participating in the consortium is increased by holobiont formation, making it apparent that a holobiont is more than the sum of its parts [1]. Moreover, according to the hologenome theory of evolution, the collective genomes of the holobiont—i.e., the hologenome—form an integrated unit of selection [2,3]. Selection will thus be acting across different time- and spatial scales on the holobiont as a whole, as well as on each individual member, with the joint holobiont fitness as the unifying trait [4].

A classic example of the additional fitness provided by the holobiont is the study by Goulet et al. [5], which compared natural and lab-produced host–symbiont genotypic combinations in a sea anemone, resulting in different net oxygen fluxes and different responses to heat stress. To understand how the holobiont functions, and to predict the effects of environmental drivers, it is therefore crucial not only to identify all of the holobiont’s partners and their individual contributions, but also to take into account the specific combinations and interactions of holobiont participants that will eventually define the genotypic and phenotypic spectrum of the holobiont.

Among the many examples of holobionts in nature, corals and lichens are arguably the most iconic. These superorganisms are the symbiotic phenotype of nutritionally specialized hosts-marine invertebrates in the case of corals, fungi in lichens—that derive fixed carbon from a population of photosynthetic symbionts—dinoflagellates in corals, green microalgae, and/or cyanobacteria in lichens. In return, the host provides micronutrients and a protected, sunlit habitat to its photosynthetic partners.

Maintaining and modulating photosynthesis under variable environmental conditions is critical for holobionts because all members of the consortium need the products of photosynthesis for nutrition and survival. Thus, resilience, robustness, acclimation, and/or adaptation of the coral and lichen holobionts ultimately depend on photosynthetic performance. Regulation of photosynthesis in holobionts may take place at several levels. In the case of corals, we know that *Symbiodinium* photobionts are not solely responsible for the photosynthetic performance of the holobiont, but the host itself has a key role in modulating photosynthesis [6,7]. Therefore, photosynthetic adaptations in corals may take place via community changes in the associated photosynthetic species, both in terms of frequency shifts and symbiont switching [8]. This may lead to light-dependent zonation of dinoflagellates along marine depth gradients [9,10,11,12]. Furthermore, the coral host contributes to holobiont fitness by providing a multitude of photophysiological and protective effects to its photosynthetic endosymbionts: It delivers enzymatic antioxidants and microsporine-like amino acids, which absorb UVR and have antioxidant activities, it controls the amount of light reaching the endosymbionts by modulating tissue thickness, and it produces a variety of fluorescent proteins, as a means to match the spectral quality of light [6]. In the case of lichens, symbiont switching is commonly considered as the most obvious similarity with corals, with several studies reporting on the ability of lichen-forming fungal hosts to associate with different, potentially locally-adapted algae as a strategy to increase the ecological amplitude of the lichen holobiont [13,14,15,16]. Furthermore, lichen individuals may carry multiple algal strains with different physiological properties, which has been interpreted as a strategy to increase ecological flexibility of the holobiont [17,18]. Interestingly, Trebouxiophyceaen photobionts in lichens have specialized reaction centers of photosystem II to prevent damages in case of abrupt changes in light intensity [19,20]. The role of the fungal host in modulating photosynthetic performance of the lichen holobiont is less understood. A few studies have suggested that the fungus may partake in the modulation of photosynthesis via (i) regulation of oxidative stress and photo-oxidative protection via nitric acid production [21], (ii) biosynthesis of melanin and other compounds to screen the photobionts from excessive UVR [22,23], (iii) regulation of algal population size, in terms of number of cells, within the lichen thallus [24,25], (iv) active positioning of algal cells beneath the peripheral cortex layer [26], and (v) regulation of thallus thickness depending on prevalent water and light conditions [27,28].

Disentangling the relative effects of fungal and algal partners on holobiont fitness is challenging in lichens, because artificial synthesis of lichen individuals with specific combinations of partners, which have different physiological properties, has so far not been possible. Alternatively, one could measure functional traits in naturally occurring fungal hosts that pair up with different algal partners in the same environment, or different fungal hosts that use the same algal symbiont in the same environment. However, this requires knowledge of the exact distribution of fungal–algal genotype pairings in nature, which is known for only a very limited number of lichen symbioses. The lack of appropriate model systems has been a limiting factor for understanding the lichen holobiont’s response to environmental cues.

Here, we investigated the role of both algal and fungal symbionts on the photosynthetic performance of a lichen holobiont. We used two species of the genus *Umbilicaria* as models: *U. pustulata* and its sister-species *U. hispanica*. This is an ideal system for disentangling host and symbiont effects on overall holobiont fitness for the following reasons: (i) both fungal host species consist of genetically distinct lineages, each with a clear high-altitude and low-altitude genotype that corresponds to climate zone [29]; moreover, in the case of *U. pustulata*, the fungal lineages represent ecotypes that are differentially adapted to climate [28]; (ii) both fungal species display consistent symbiont turnovers along climatic gradients, and share the same algal symbiont strains [16,29]; (iii) all fungal-algal pairings may occur sympatrically in the transition zone between climates across mountains in the Mediterranean region [28,30]. Based on this pre-existing knowledge on the mosaic of symbiont combinations in these species, we addressed the following questions: (1) Do different algae affect photosynthetic performance of a lichen thallus? (2) Do different fungal species and/or different fungal genotypes affect photosynthetic performance of a lichen thallus? For this, we compared chlorophyll fluorescence kinetics between the two sister species and among lichen thalli representing various genotypic combinations collected from the same local population.

## 2. Materials and Methods

### 2.1. Sampling and Detection of Fungal-Algal Genotypic Combinations

Whole lichen thalli (5–8 cm in diameter) were collected in the Sierra de Gredos mountain range (Sistema Central, Spain) at an elevation of 1500 m a.s.l. (Spain, Ávila, Sierra de Gredos, Puerto de Candeleda, on silicate rocks along a hiking trail; latitude, longitude: 40.22384, −5.23869). Recent studies investigated the fungal population structure based on genome-wide SNP analysis [29] and algal community diversity and turnover based on ITS2 metabarcoding [30]. Results from these studies have shown that the *U. pustulata* and *U. hispanica* populations associate predominantly with the algae *Trebouxia* OTU1 and OTU2 [30] at this elevation. At this elevation, we find only the cold adapted fungus of *U. pustulata*
*sensu* [28], whereas we find both genetic lineages of *U. hispanica* (H3: low elevation; H1: high elevation; *sensu* [30]). To identify these intraspecific fungal lineages, we genotyped each sample (whole lichen thalli) at the fungal single-copy protein coding gene *MCM7*. For the green algal symbionts, we identified the algal species using the internal transcribed spacer region (ITS) of the rRNA operon. Each sample was genotyped following Sadowska-Deś et al. [31]. Based on the observed fungal–algal pairing, we considered five genetic fungal–algal combination (groups) for photosynthetic analysis: Group 1: *U. hispanica*, fungal H3, algal OTU1; Group 2: *U. hispanica*, fungal H1, algal OTU1; Group 3: *U. hispanica*, fungal H3, algal OTU2; Group 4: *U. pustulata*, fungal P1, algal OTU1; Group 5: *U. pustulata* fungal P1, algal OTU2. We made the following comparisons: different fungal host genotype, same algal symbiont (Group 1 vs. Group 2); different fungal host species, same algal symbiont (Group 2 vs. Group 4, and Group 3 vs. Group 5); different algal symbiont, same fungal host (Group 1 vs. Group 3, and Group 4 vs. Group 5). Sample information and GenBank accession numbers are given in Table 1.

### 2.2. Analysis of Photosynthetic Performance

Frozen (−20 °C) lichen samples were thawed and acclimatized prior to photosynthetic performance analysis. For this, we submerged the lichen thalli in water and placed it in a Petri dish lined with water-soaked three-layered filter paper. These Petri dishes were then kept in a plant chamber (CFL Plant Climatics) for 72 h, with a day/night cycle of 12 h, 16 °C, and light intensity of 33 μE. During these 72 h, humidity was kept constant and condensed water was removed. The chlorophyll a fluorescence measurements were performed by Phytoprove Pflanzenanalytik UG.

To analyze differences in photosynthetic performance of each fungal–algal pairing, we used chlorophyll a fluorescence rise kinetics (O-J-I-P transients) [32]. In photosynthetic organisms, fluorescence transients rise from F0 (the state when all PSII reaction centers are open, i.e., the primary acceptor quinone is fully oxidized) to FM (the state when all the PSII reaction are closed, i.e., the primary acceptor quinone is fully reduced). The polyphasic transients show different steps, including O for origin (corresponding to F0), J and I for intermediate levels (J at 2 ms, I at 30 ms), and P for the peak (corresponding to FM) [33]. To perform the analysis, we kept the lichen thalli in the dark for one h. For each sample, we measured chlA fluorescence rise kinetics at room temperature (20 °C) using a Plant Efficiency Analyzer (Pocket PEA fluorimeter, manufactured by Hansatech Instruments Ltd., King’s Lynn, UK) with Hansatech leaf clips covering a circular area of 2 mm diameter on the lichen thallus. The emitter wavelength of a non-modulated light source was 625 nm for the actinic light LED and the saturating light intensity for the measurements was 3.500 μmol photons m^2^ s^−1^. Raw data were transferred and processed using PEA Plus software (Hansatech Instruments Ltd., King’s Lynn, UK). For analyzing differences in the curve shape, chlA fluorescence induction curves were normalized to F0 = 50 µs. The data were plotted graphically by using “Prism 8 for Mac OS-X” software (GraphPad Software Inc., La Jolla, USA). The same software was used to determine significant differences (*p* < 0.05) in the different steps of the chlA fluorescence induction curves between the different groups. For this, unpaired t-tests with Welsh’s correction were performed and the F-Test to compare variances. To detect variances within the lichen thalli, three replicates per lichen thallus were analyzed at different sites of each thallus. The area of each measuring point was 0.283 cm^2^. Statistical analyzes were performed using GraphPad Prism 9.0 La Jolla, USA. All calculated parameters are relative parameters formed by quotients and are thus independent of the amount of chlorophyll within the photobionts.

## 3. Results

We analyzed chlA fluorescence rise kinetics for four different fungal–algal genotype combinations (Figure 1). This study design allowed us to test the algal effect on photosynthesis by comparing the photosynthetic performance between different algal species associated with the same fungal genotype. Additionally, we could test the effect of the fungal genetic background on photosynthesis by comparing the photosynthetic performance between different fungal species and different intraspecific fungal genotypes associated with the same algal species.

Different algae associated with the same fungal haplotype of *U. pustulata* (P1) displayed significantly different photosynthetic performance, with algal OTU1 performing better than algal OTU2 under the given experimental conditions (Figure 1A, Appendix A). A similar comparison of algal performance in *U. hispanica* did not yield significant differences (*p*-value J-step: 0.06, *p*-value I-step: 0.1, adjusted *p*-value of the ANOVA = 0.17, Appendix A). The performance of the individual algal OTUs was also dependent on the host species: comparing the photosynthetic performance of the same alga associated with different fungal host species, the performance of *Trebouxia* OTU1 (Figure 1B) as well as OTU2 (Figure 1C) in *U. hispanica* was significantly different (for both, higher) than in *U. pustulata*. Photosynthetic performance was also modulated by an intraspecific host genetic background: the pairing algal OTU1-*U. hispanica* H1 performed significantly different from the pairing algal OTU1-*U. hispanica* H3 (Figure 1D).

## 4. Discussion

Lichens are widespread terrestrial symbioses consisting of nutritionally specialized fungi that associate with green algae and/or cyanobacteria to form self-sustained holobionts. Photosynthesis is therefore a key physiological process in lichens. The coordinated regulation of photosynthesis between symbionts plays a crucial role in optimizing the fitness of the lichen holobiont. The example of the common orange lichen, *Xanthoria parietina*, illustrates the interplay of fungal and algal physiological processes in photosynthesis regulation: The fungal orange pigment parietin protects the photosynthetic apparatus of algal symbionts against damage by high light levels [22]. In the algal symbiont, the protection of the photosynthetic apparatus is further modulated by nitric oxide [34,35,36]. Biosynthesis of parietin (by the fungus) is induced by UV radiation [37], but it can be further stimulated and accelerated by high concentrations of the algal photosynthate ribitol [38]. These findings highlight the closely intertwined physiological responses of fungal host and algal symbionts to environmental cues. In the present study, we studied the individual contribution of the eukaryotic components of the lichen holobiont to photosynthetic performance by assessing the photosynthetic performance of lichen thalli of known fungal–algal genotype combinations.

### 4.1. Photobiont Switches Are Key to a Plastic Holobiont Response

Our data confirm the results from previous studies that different algae can display different photosynthetic performance. Casano et al. [17] showed that two common photobionts of the lichen *Ramalina farinacea*, often co-inhabiting the lichen thallus, have clearly distinct physiological optima. The authors suggested that these physiological differences might be a key factor in increasing the niche space of the lichen. Indirect confirmation of this hypothesis comes from studies on the biogeography of fungal–algal associations in different lichens, which show predictable switches to different photobiont species over broad latitudinal and/or elevational gradients, and climatic niche preferences of some of the photobionts [14,16,30,39].

### 4.2. The Host Effect on Photosynthetic Performance

Similarly to corals, our results show that the host of the symbiosis—the fungus in the case of lichens—affects photosynthetic performance. Additionally, we show—to our knowledge for the first time—that this host effect on photosynthesis is not only a species-specific phenomenon, but that it is also affected by the genetic diversity within a host, with different intraspecific lineages of a fungus causing different responses.

The fact that different fungi may utilize the same algae in different ways is not surprising, if one considers the differences in thallus morphologies and chemical components in lichens. Associating with different fungal hosts might be a way for a given alga to increase its niche space and colonize different habitats [40]. The result of this process is the establishment of articulated fungal networks centered on the horizontal sharing of a common photobiont, the so-called photobiont-mediated guilds [15,41]. Photobiont-mediated guilds might be more common in nature than currently recognized [42].

The finding that different intraspecific lineages within the same fungal host may lead to different photosynthetic responses sheds new light on the potential functional role of the cryptic genetic diversity that is commonly reported for many lichen-forming fungal species. The mechanisms behind this photosynthetic modulation by the fungus are yet unclear. It has been reported that different *Cetraria aculeata* mycobionts may acclimate to different environments by regulating algal cell numbers (chlorophyll content) within the thallus [25]. The regulation of thallus thickness may also play a role in the acclimation of the lichen holobiont, as shown for *U. pustulata* fungal ecotypes inhabiting different climates [28]. Acclimation in lichens is also mediated by differential chemical composition of the thallus under different environmental conditions [43], which is supported by the observation of UV-induced biosynthesis of anthraquinones and melanins [38]. Further support to this hypothesis comes from a recent study showing that *U. pustulata* fungal ecotypes from different climates differ in their secondary biosynthetic potential, based on genome-wide, population level comparisons of biosynthetic genes [44]. To conclude, the correct interpretation of host fungal modulation of the lichen’s photosynthesis must be based on a polyphasic approach aimed at characterizing anatomical features, secondary metabolite profile, structure of the cortex, and mycobiont genetic diversity.

### 4.3. Assembling the Holobiont Puzzle: Conservation Outcomes and Future Perspectives

According to the holobiont concept, the response of a holobiont to environmental cues is a complex trait derived from multiple interactions among its bionts. Therefore, investigating the biodiversity of the holobiont components is an essential step in understanding the functional outcome of the assembled holobiont, and in predicting its responses to environmental change. Recently, it has been shown in corals that thermal tolerance is the outcome of an integrated adaptive response of the different constituents of the coral holobiont, i.e., coral host, algal symbiont, and associated microbiome [45]. According to this, the differential fitness of symbiotic associations across a range of ecological settings may determine the holobiont’s success and therefore the frequency of any given association in a particular environment. Applied to the lichen model, the environmentally determined success of a given association will therefore be the key driver in determining the frequency of that association at a site. This idea has been implemented to explain the variation in associations and the varied selectivity exhibited by mycobionts towards different algal strains in lichens [16,46].

Our study builds on this idea and broadens its functional significance. We showed that both algae and fungal genotypes and/or species contribute uniquely to the resulting photosynthetic performance in lichens. Lichens are often used as examples of holobionts with very broad distributions and tolerances across wide ecological ranges [47]. However, our findings indicate that this might be the result of an optimization of responses, where different fungal–algal pairs provide different, probably environment-specific responses.

Based on these findings, we are able to highlight two key vulnerabilities of the lichen holobiont to environmental change. Firstly, if the modulation of responses of a common alga shared among members of a photobiont-mediated guild may facilitate the dispersal and establishment of the fungi in the guild, this interdependence may have catastrophic effects on the overall fungal community in case the common photobiont was lost. Secondly, as regional and global environments change, the ecological outcomes of fungal–algal interactions can also change, potentially causing the loss of unique, environmentally-tuned pieces of the holobiont puzzle. The resulting fragmentation of the holobiont would lower its capacity to respond and adapt to further changes. As a result, lichens—examples of stress-tolerant organisms—might be inherently more vulnerable to changes than currently recognized.

## 5. Conclusions

In conclusion, our study highlights the importance of including a holistic perspective in studying the lichen holobiont. Future studies should focus on identifying the different genotypic combinations that constitute a lichen species—possibly also including its prokaryotic component—and how their differential responses influence the holobiont properties. Only by understanding how its individual components interact with each other as well as their functional outcome will allow us to predict the response to climate change of the lichen holobiont as a whole.

## Figures and Tables

**Figure 1 jof-08-01267-f001:**
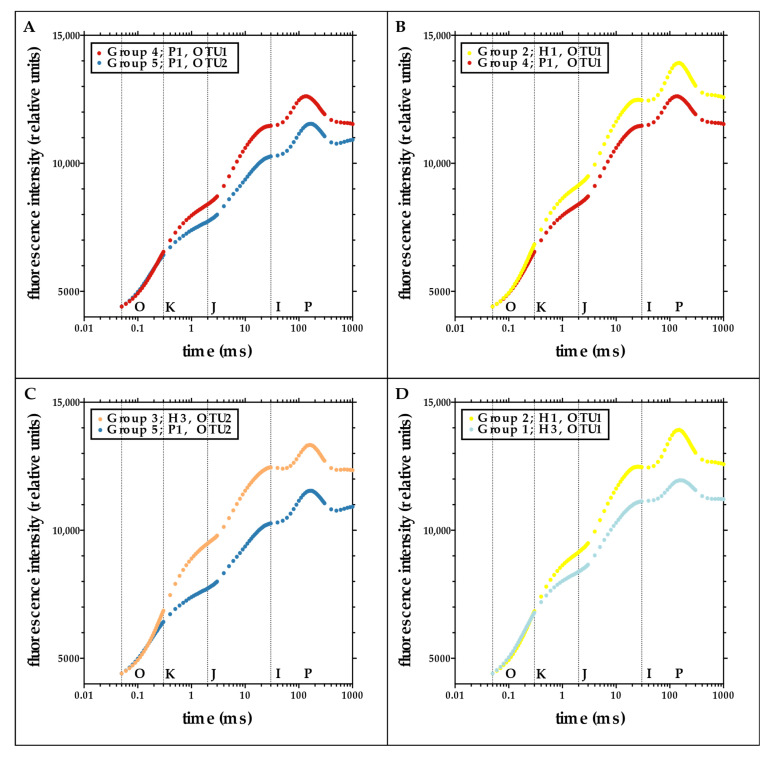
Effect of different algae (**A**), different fungal species (**B**,**C**), and different intraspecific fungal genotypes (**D**) on photosynthetic performance in lichens based on chlorophyll a fluorescence rise kinetics (O-J-I-P-transients). This method displays differences in the effectiveness of photosystems in reducing the various plastoquinones along the electron transport chain using the maximal impact of light from a dark-adapted state. We performed the following comparisons: (**A**) Different algal OTU, same fungal genotype: group 4 (red) vs. group 5 (dark blue). (**B**,**C**) different fungal species, same algal OTU: group 2 (yellow) vs. group 4 (red) and group 3 (orange) vs. group 5 (dark blue). (**D**) Different intraspecific fungal genotypes, same algal OTU: group 2 (yellow) vs. group 1 (light blue). Further information on samples in each group is given in Table 1. Analysis based on a *t*-test with Welsh’s correction. All differences in the O-J-I-P steps of the chlorophyll fluorescence induction curves were significant in this graph (see Appendix A: Fluorescence intensities for the J-, I-, and P-step for the comparisons of the chlorophyll a fluorescence rise kinetics (O-J-I-P-transients) for Figure 1).

**Table 1 jof-08-01267-t001:** Specimens used in this study. Samples are grouped into five groups according to their specific fungal–algal combinations. Designation of fungal haplotypes and algal OTUs follows [30]. Samples are deposited in the Herbarium Senckenbergianum Frankfurt (FR).

Group	Sample Number/ Herbarium Number	Fungal Species	Fungal Haplotype	GenBank Accession (*MCM7*)	Algal Strain (OTU)	GenBank Accession (ITS)
Group 1	5314	*U. hispanica*	H3	OP834011	OTU1	OP852400
	5318			OP834012		OP852401
	5329			OP834013		OP852402
	5334			OP834014		OP852403
Group 2	5316	*U. hispanica*	H1	OP834015	OTU1	OP852404
	5319			OP834016		OP852405
	5327			OP834017		OP852406
	5331			OP834018		OP852407
Group 3	5324	*U. hispanica*	H3	OP834019	OTU2	OP852408
	5330			OP834020		OP852409
Group 4	5336	*U. pustulata*	P1	OP834021	OTU1	OP852410
	5337			OP834022		OP852411
	5351			OP834023		OP852412
	5352			OP834024		OP852413
	5354			OP834025		OP852414
	5355			OP834026		OP852415
Group 5	5335	*U. pustulata*	P1	OP834027	OTU2	OP852416
	5344			OP834028		OP852417
	5359			OP834029		OP852418
	5360			OP834030		OP852419

## Data Availability

The data presented in this study are available in NCBI GenBank.

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
