# Peer review of "Fungal Host Affects Photosynthesis in a Lichen Holobiont"

_jof, 2022, doi:10.3390/jof8121267_

Round 1

Reviewer 1 Report

The authors have studied if the Genetic background of fungal host affects photosynthesis in two lichen holobionts of the lichenized genus Umbilicaria.

     The manuscript is very synthetic and well written. The introduction is well developed, although it does not always contain the most relevant or up-to-date bibliography on some of the physiological or genetic aspects of the lichen symbioses referenced.

     The methodology is adequate for the stated objectives. The results are consistent with the methods used, although it is not well understood why they refer to their 2018 metabarcoding results ( ) and then do not take into consideration that some of the thalli used for this work may have more microalgae in addition to Trebouxia OTU1 and OTU2. The discussion is well presented although it suffers from what was said for the introduction, there is a lack of references (not only self-citations) that would illustrate more clearly the conclusions obtained.

     In any case, it is a good paper and I believe it can be accepted for publication in this journal, once the authors have refined the contents. 

I make a small compilation of suggestions:

     1. Abstract: line 21: can be regarded AS a response to the environment.

     2. Our findings suggest that characterizing the genetic composition of both eukaryotic partners is an essential step to understand and predict the lichen holobiont's responses to environmental change>>

I ask: Would a different adjective be more appropriate? Since anatomy and other variables may be just as relevant?

     3. Update and expand the bibliography of the concepts presented between lines 49-79; 217-240, And also in the discussion, e.g. special functioning of photosystem II in these microalgae or the alterations of parameters of fluorescence kinetics modulated by water deficit or by pollutants, or the role of nitric oxide in the protection of the photosynthetic apparatus, etc.

     4. In material and methods, please clarify:

          4.a.How long the lichens used for the experiments were kept in chambers before acclimation.

          4.b. They state that these two Umbilicaria species are predominantly associated with the microalgae OTU1 and OTU2, but in the previous papers on which they are based, they refer to predominant microalgae but with the concurrence (metabarcoding) of other Trebouxia species.

          4.c. They used 8 mm thallus pieces to measure chlorophyll a fluorescence. Please clarify: i) size of thalli used (it has been shown that thallus size can exhibit different microalgal guilds in the same species and habitat. ii) thallus areas used in fluorescence measurements.

     5. Clarify, for example, that the correct interpretation of host fungal modulation must also be based on the anatomy, the composition of secondary metabolic substances and/or the structure of the cortex.

  • Deep-coverage Illumina DNA metabarcoding was used to track changes in the community composition of Trebouxia algae associated with two phylogenetically closely related, but ecologically divergent fungal hosts along a steep altitudinal gradient in the Mediterranean region.

An understanding of how biotic interactions shape species’ distributions is central to predicting host–symbiont responses under climate change. Switches to locally adapted algae have been proposed to be an adaptive strategy of lichen-forming fungi to cope with environmental change. However, it is unclear how lichen photobionts respond to environmental gradients, and whether they play a role in determining the fungal host's upper and lower elevational limits.

  • We detected the presence of multiple Trebouxia species in the majority of thalli. Both altitude and host genetic identity were strong predictors of photobiont community assembly in these two species. The predominantly clonally dispersing fungus showed stronger altitudinal structuring of photobiont communities than the sexually reproducing host. Elevation ranges of the host were not limited by the lack of compatible photobionts.
  • Our study sheds light on the processes guiding the formation and distribution of specific fungal–algal combinations in the lichen symbiosis. The effect of environmental filtering acting on both symbiotic partners appears to shape the distribution of lichens.

Author Response

We thank the Reviewer for the frank and positive evaluation of our manuscript. We addressed all the points that were raised. Specifically, we have expanded the bibliography section on photosynthetic kinetics and added the requested details on the experiments.

I make a small compilation of suggestions:

  1. Abstract: line 21: can be regarded AS a response to the environment.

Done.

  1. Our findings suggest that characterizing the genetic composition of both eukaryotic partners is an essential step to understand and predict the lichen holobiont's responses to environmental change>>

I ask: Would a different adjective be more appropriate? Since anatomy and other variables may be just as relevant? 

We have changed the sentence as follows: “Our findings suggest that characterizing the genetic composition of both eukaryotic partners is an important complimentary step to understand and predict the lichen holobiont's responses to environmental change”.

  1. Update and expand the bibliography of the concepts presented between lines 49-79; 217-240, And also in the discussion, e.g. special functioning of photosystem II in these microalgae [19] or the alterations of parameters of fluorescence kinetics modulated by water deficit or by pollutants [20], or the role of nitric oxide in the protection of the photosynthetic apparatus [34-36], etc.

We thank the Reviewer for the suggestions. We have expanded and updated the bibliography accordingly:

Intro: “Interestingly, Trebouxiophyceaen photobionts in lichens have specialised reaction centers of photosystem II to prevent damages in case of abrupt changes in light intensity [19].”

Discussion: … against damage by high light levels [20]. In the algal symbiont, the protection of the photosynthetic apparatus is further modulated by nitric oxide [34-36].”

  1. In material and methods, please clarify:

          4.a.How long the lichens used for the experiments were kept in chambers before acclimation.

We have added details in the Materials and Methods section as follows: “Frozen (-20 °C) lichen samples were thawed and acclimatized prior to photosynthetic performance analysis. For this, we submerged the lichen thalli in water and placed it in a Petri dish lined with water-soaked three-layered filter paper. These Petri dishes were then kept in a plant chamber (CFL Plant Climatics) for 72 hours, with a day/night cycle of 12 hours, 16°C and light intensity of 33 μE. During these 72 hours, humidity was kept constant and condensed water was removed”.

          4.b. They state that these two Umbilicaria species are predominantly associated with the microalgae OTU1 and OTU2, but in the previous papers on which they are based, they refer to predominant microalgae but with the concurrence (metabarcoding) of other Trebouxia species.

According to our previous work on the topic (Paul et al. 2018), Sanger sequencing works well when there is a single predominant photobiont. In this study we therefore performed the physiological experiments on thalli that gave clear Sanger electropherograms, thus containing a single predominant alga.

          4.c. They used 8 mm thallus pieces to measure chlorophyll a fluorescence. Please clarify: i) size of thalli used (it has been shown that thallus size can exhibit different microalgal guilds in the same species and habitat. ii) thallus areas used in fluorescence measurements.

We apologise for the confusion. The correct statement is that we used whole thalli of ~5-8 cm in diameter. We chose this thallus-size range to partially control for and minimize the differences among samples thus making the results comparable. Three fluorescence measurements were taken in different parts of each thallus.

As for the thallus areas used in fluorescence measurements, the area of each measuring point (3 per thallus) was 0.283 cm2.

We updated the text accordingly.

“Whole lichen thalli (5-8 cm in diameter) were collected in the Sierra de Gredos mountain range (Sistema Central, Spain) at an elevation of 1500 m a.s.l.”

“To detect variances within the lichen thalli, three replicates per lichen thallus were analyzed at different sites of each thallus. The area of each measuring point was 0.283 cm2.

  1. Clarify, for example, that the correct interpretation of host fungal modulation must also be based on the anatomy, the composition of secondary metabolic substances and/or the structure of the cortex.

The following sentence was added to the Discussion section: “To conclude, the correct interpretation of host fungal modulation of the lichen’s photosynthesis must be based on a polyphasic approach aimed at characterizing anatomical features, secondary metabolite profile, structure of the cortex and mycobiont genetic diversity.”

Author Response

We thank the Reviewer for the thorough analysis of possible myco-photobiont pairings and for the question. Basically, our analysis has been limited by the availability of thalli and genotypic fungal-algal combinations in nature. We chose to analyze a single population in order to 1) minimize potentially confounding effects derived from local, site-dependent environmental conditions, 2) maximize the comparability of photosynthetic measurements while ensuring at the same time that were did not wipe out the entire lichen population (for the ecophysiological analysis we sampled entire thalli). As for the first point, we selected the population at 1500 m a.s.l. as, in our previous article (Ref [28]), it was shown to harbor the highest number of different fungal-algal combinations. The photobiont P3 does not occur at that elevation. A further restriction came from the fact that we tried to maximize comparability by sampling thalli of similar size (~5-8 cm in diameter) without knowing a priori the relative abundance of the different fungal-algal combinations. This further reduced the number of groups that could be used for comparison as some of the combinations were underrepresented according to the genotypic analysis.

By the way, names of two common authors in this manuscript and Ref [28] are differently listed: This ms: Schmitt Imke, Dal Grande Francesco;
Ref [28]: Imke Schmitt, Francesco Dal Grande

The information was rectified.

Overall, the study is well designed, and the manuscript is well prepared. However, as I point out in the Specific Comments, there is an important disagreement in the results that may affect the conclusion. The disagreement should not be ignored, and therefore the manuscript should be revised accordingly.

Thank you for the positive feedback and for raising a point that certainly deserves to be more clearly addressed and clarified in the manuscript (more about this below). 

Specific Comments
Title may better be changed as “Genetic background ...” à “Genotypic variation ...” because no elevation-related genes were characterized.

Done.

Table 1 should accommodate GenBank accession numbers when they are available.

Yes. We have submitted the sequences to GenBank and the accession numbers will be added to the Table by the time of acceptance.

Figure 1 should show Group names directly on each panel (panels A–D), as it is hard to interpret the results from the current panels. Panel A shows the comparison between Groups 4 and 5 that indicates higher photosynthetic performance of the alga OTU1 than OTU2. In Panels B, the alga OTU1 showed higher performance in the fungus H1 (elevation level ≧V, high) than in P4 (elevation level ≦III, low). Similarly in Panel C, the alga OTU2 showed higher performance in the fungus H1 than in P4, too. And, Panel D shows higher performance of the alga OTU1 in the fungus H1 than in H3 (elevation level ≦V, rather low).

Comparison of the Groups 1 and 3 (same fungus / different algae), that is, the fungus H3 with the alga OTU1 or OTU2, is missing. The comparison is done with the light blue (Group 1 = H3 + OTU1) in Panel D and the orange (Group 3 = H3 + OTU2) in Panel C and shows that the alga OTU2 performs better than OTU1, which disagrees with the result from Panel A and the statement “algal OTU1 performing better than algal OTU2” (L177–178). Please point out the disagreement explicitly in the text and give it an explanation and discussion.

I will resume reviewing when the manuscript is appropriately revised, particularly in terms of the disagreement mentioned above.

We thank the Reviewer for suggesting adding the Group names directly onto each panel of Fig. 1. We concur with the Reviewer that this addition adds further clarity.

As for the suspected disagreement in the reports on algal performance, we have also checked the comparison indicated by the Reviewer, namely ‘H3 + OTU1’ vs ‘H3 + OTU2’, and there were no statistically significant differences between the groups (adjusted p-value of the ANOVA = 0,1653). We believe that the lack of significance is simply an indication that we did not have enough biological replicates (and thus data) for one of the groups in the given comparison (6 measurements for 2 samples of ‘H3 + OTU2’).

As pointed out by the Reviewer, the plots suggest a better performance of OTU2 in U. hispanica. This, however, not only is NOT in disagreement with our statement that OTU1 performs better than OTU2 in its sister-species, U. pustulata, but also is an additional confirmation that the same alga may behave differently when associated with different fungi. We refrained from reporting the H3 + OTU1’ vs ‘H3 + OTU2’ due to the lack of statistical support for this comparison, instead we mention in the Results section that “A similar comparison of algal performance in U. hispanica did not yield significant differences (adjusted p-value of the ANOVA = 0.17; data not shown).”

Round 2

Author Response

We thank the reviewer for the second revision and his comments. We appreciate the suggestions and added a table in the supplementary (Table S1) to give all the ANOVA adjusted p-value correlations of the J- and the I-step. Moreover, we agree that the two sentences were duplicates and removed it in the updated manuscript version. Concerning the last question, yes, we confirm that the comparison was not statistically significant (p-value J-step: 0,06, p-value I-step: 0.1, adjusted p-value of the ANOVA = 0.17, Table S1).